# LLaVA-4D: Embedding SpatioTemporal Prompt into LMMs for 4D Scene Understanding

**Hanyu Zhou**[1], **Gim Hee Lee**[1]

[1] School of Computing, National University of Singapore

`{hy.zhou, gimhee.lee}@nus.edu.sg`

## ABSTRACT

Despite achieving significant progress in 2D image understanding, large multimodal models (LMMs) struggle in the physical world due to the lack of spatial representation. Typically, existing 3D LMMs mainly embed 3D positions as fixed spatial prompts within visual features to represent the scene. However, these methods are limited to understanding the static background and fail to capture temporally varying dynamic objects. In this work, we propose LLaVA-4D, a general LMM framework with a novel spatiotemporal prompt for visual representation in 4D scene understanding. The spatiotemporal prompt is generated by encoding 3D position and 1D time into a dynamic-aware 4D coordinate embedding. Moreover, we demonstrate that spatial and temporal components disentangled from visual features are more effective in distinguishing the background from objects. This motivates embedding the 4D spatiotemporal prompt into these features to enhance the dynamic scene representation. By aligning visual spatiotemporal embeddings with language embeddings, LMMs gain the ability to understand both spatial and temporal characteristics of static background and dynamic objects in the physical world. Additionally, we construct a 4D vision-language dataset with spatiotemporal coordinate annotations for instruction fine-tuning LMMs. Extensive experiments have been conducted to demonstrate the superiority of our method on various tasks of 4D scene understanding. Our code: *https://github.com/hyzhouboy/LLaVA-4D.*

## 1 INTRODUCTION

Large multimodal models (LMMs) (Alayrac et al., 2022; Liu et al., 2023) aim to learn the representation alignment between language and other modalities such as vision and audio. They have been widely applied in multiple scene understanding tasks such as dense caption (Wang et al., 2022), visual QA (Li et al., 2022a; 2023), scene grounding (Kamath et al., 2021), *etc*. Although recent language-vision LMMs including LLaVA (Liu et al., 2023) and PaLI (Chen et al., 2023b) have achieved great success in 2D image understanding, they still face challenges in the 3D physical world. This is because these LMMs trained solely on 2D images lack the representation of 3D spatial characteristics to interact with the physical world. In this work, our purpose is to improve the characteristic representation and scene understanding of LMMs for the physical world.

As shown in Fig. 1(a), existing 3D language-vision LMM methods (Hong et al., 2023; Zhu et al., 2024a;a; Zheng et al., 2024; Chen et al., 2024a) use 3D positions as spatial prompts which are then embedded into visual features to represent the whole scene. For example, Hong et al. (2023) extract 2D visual features from multi-view images and use 3D positions to transform these features into corresponding 3D inputs for LMMs. However, these 3D LMM methods can only handle the static background with limited ability to understand dynamic objects in the scene. Unlike static backgrounds, dynamic objects exhibit temporally varying spatial characteristics such as position shifts and deformations. Unfortunately, existing 3D LMM frameworks neglect temporal aspects in using a unified spatial representation for the entire scene. As shown in Fig. 1(c-d), 3D LMMs perform poorly on dynamic understanding tasks. This highlights the need to capture both spatial and temporal information to improve scene understanding in dynamic physical environments.

As shown in Fig. 1, we propose a novel spatiotemporal prompt embedded with the visual representation to model the spatiotemporal characteristics of the scene. We design the spatiotemporal

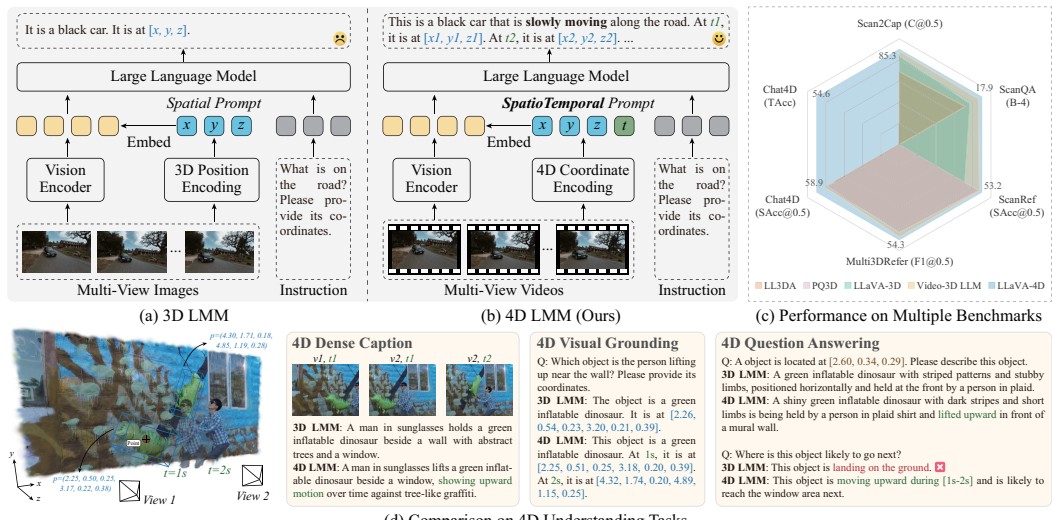

Figure 1: Illustration of 3D and 4D LMM paradigms for physical world understanding. (a) Existing 3D-LMMs encode 3D positions as spatial prompts but overlook dynamic objects. (b) Our 4D LMM framework embeds 4D coordinates: position and time as spatiotemporal prompts to capture both background and dynamic objects. (c) Performance comparison of LMMs on benchmarks. (d) Comparison on 4D understanding tasks.

prompt based on the observation that dynamic objects and static backgrounds share similar 3D positional encoding but differ significantly in motion patterns. This motivates the extension of 3D positional encoding into a dynamic-aware 4D coordinate encoding to better differentiate objects from the background. Additionally, we observe that visual features of a scene can be disentangled into spatial and temporal components, which effectively distinguish objects from the background. This inspires the design of a spatiotemporal-disentangled vision embedding for scene representation. The 4D spatiotemporal prompt and visual spatiotemporal features jointly model the fine-grained spatiotemporal characteristics of dynamic objects and static backgrounds, thereby enhancing the 4D scene understanding of the physical world by LMMs.

In this paper, we propose LLaVA-4D, a general language-vision large multimodal model for 4D scene understanding. As illustrated in Fig. 2, our LLaVA-4D includes a dynamic-aware 4D coordinate encoding and a spatiotemporal-disentangled vision embedding. The 4D coordinate encoding module constructs 4D coordinates for multi-view videos and incorporates optical flow to enhance spatiotemporal encoding. The vision embedding module disentangles visual features from multi-view videos into spatial and temporal components, and enriches these spatiotemporal features with dynamic-aware 4D coordinate embeddings through cross-attention fusion. We further perform spatiotemporal encoding on textual 4D coordinates within language embeddings, which are then aligned with the fused visual spatiotemporal embeddings. In this unified framework, 4D coordinate encoding and visual spatiotemporal embedding collaboratively enhance the modeling of dynamic and static elements to improve 4D scene understanding in LMMs. Additionally, we present *Chat4D*, a 4D vision-language dataset with spatiotemporal coordinate annotations designed to instruction-tune our model for more effective 4D scene understanding. Our main contributions are summarized as follows:

- We are the first to propose a general vision-language large multimodal model for 4D scene understanding. Our model embeds a 4D spatiotemporal prompt into visual representation to enable LMMs to comprehend both dynamic objects and static backgrounds.

- We observe that backgrounds and objects share similar 3D spatial position encoding but exhibit distinct motion patterns in the temporal dimension. This motivates the design of a dynamic-aware 4D coordinate encoding as a spatiotemporal prompt to distinguish objects from backgrounds.

- We discover that spatial and temporal components disentangled from visual features are more discriminative for background and objects. This inspires the spatiotemporal-disentangled vision embedding for scene representation.

- We build a 4D vision-language dataset with coordinate annotations for instruction fine-tuning and conduct extensive experiments to verify the effectiveness of our method in 4D scene understanding.

## 2 RELATED WORK

**2D Vision-Language LMMs.** Leveraging the strong reasoning capabilities of large language models (Brown et al., 2020; Touvron et al., 2023a;b), numerous vision-language LMMs (Alayrac et al., 2022; Chen et al., 2023a; Li et al., 2023; Lin et al., 2024; Liu et al., 2023; 2024; Li et al., 2025a) have been developed to learn the correspondence between image-based visual and linguistic representations with broad applications in 2D scene understanding tasks. However, these vision-language LMM methods cannot work well when applied to the 3D physical world. This is because these LMMs lack the representation of 3D spatial characteristics and can only learn visual knowledge within the camera plane from 2D images, leading to underperformance of LMMs on 3D scene understanding. We thus aim to enhance the visual representation and improve LMMs on 3D scene understanding.

**3D Vision-Language LMMs.** The main challenge in understanding the physical world is representing 3D spatial characteristics. Some researchers (Hong et al., 2023; Wang et al., 2023b; Chen et al., 2024a; Zhu et al., 2024a; Deng et al., 2025; Zhi et al., 2024; Zheng et al., 2024) address this by embedding 3D positions as spatial prompts within visual features, using point-based or image-based approaches. Point-based methods (Wang et al., 2023b; Chen et al., 2024a; Zhi et al., 2024; Deng et al., 2025) reconstruct point clouds from 3D positions and use a 3D vision encoder to extract features for LMMs. To reduce reliance on reconstruction precision (Kerbl et al., 2023), image-based methods (Zhu et al., 2024a; Hong et al., 2023; Zheng et al., 2024) encode multi-view images into 2D visual features concatenated with embeddings of 3D positions. However, these methods use unified spatial representations that limit their ability to capture dynamic objects with temporal variations. We thus propose a novel spatiotemporal prompt for dynamic scene representation, and embed it into multi-view visual features to enable 4D scene understanding in LMMs.

**4D Vision & Language.** A related line of research is 4D vision and language (Li et al., 2025b; Deng et al., 2024; Sun et al., 2024), which focuses on modeling spatiotemporal characteristics. Li et al. (2025b) use 4D Gaussians (Wu et al., 2024) to represent dynamic scenes for semantic caption queries of different targets. Deng et al. (2024) introduce a 4D encoder to directly extract scene visual features for alignment with object recognition texts. However, these 4D models have two limitations. First, they are usually task-specific and can only handle similar cases within the same data distribution of training set. Second, they adopt the same representation strategy for dynamic objects and static background, which has the potential risk of misalignment of heterogeneous features. In this work, we present the first general LMM for different tasks of 4D scene understanding by disentangling visual features and embedding 4D spatiotemporal prompts to differentiate objects and background.

## 3 OUR LLAVA-4D

**Overview.** Fig. 2 shows the architecture of our LLaVA-4D. Given a multi-view video input sequence $I$, our LLaVA-4D achieves 4D scene understanding progressively through the following three stages:

1) **Dynamic-Aware 4D Coordinate Encoding (*cf.* Sec. 3.1).** This is the 4D prompt construction stage where we construct 4D coordinate tensors $[x, y, z, t]$ from multi-view videos using visual geometry, and perform spatiotemporal encoding $\text{PE}(\cdot)$, $\text{TE}(\cdot)$ on the coordinates. The encoded position and time are concatenated as a spatiotemporal prompt to guide visual fusion, *i.e.*:

$$p_{4D} = w_p \cdot [\text{PE}(x, y, z) \parallel \text{TE}(t) \cdot \beta], \tag{1}$$

where $w_p$ is MLP-based learnable parameter and $\beta$ is optical flow for temporal dynamic awareness.

2) **Spatiotemporal-Disentangled Vision Embedding (*cf.* Sec. 3.2).** This is the visual representation stage where we extract visual features $f$ from multi-view videos using a vision encoder, and disentangle these visual features into spatiotemporal components:

$$f_s, f_t = \text{STD}(f), \tag{2}$$

where $f_s$ is spatial feature and $f_t$ is temporal feature. We further embed encoded 4D coordinate features into these spatiotemporal features via cross-attention fusion:

$$f_{st} = \text{CAtt}([f_s, f_t], p_{4D}), \tag{3}$$

where $f_{st}$ denotes the output visual spatiotemporal feature with 4D awareness.

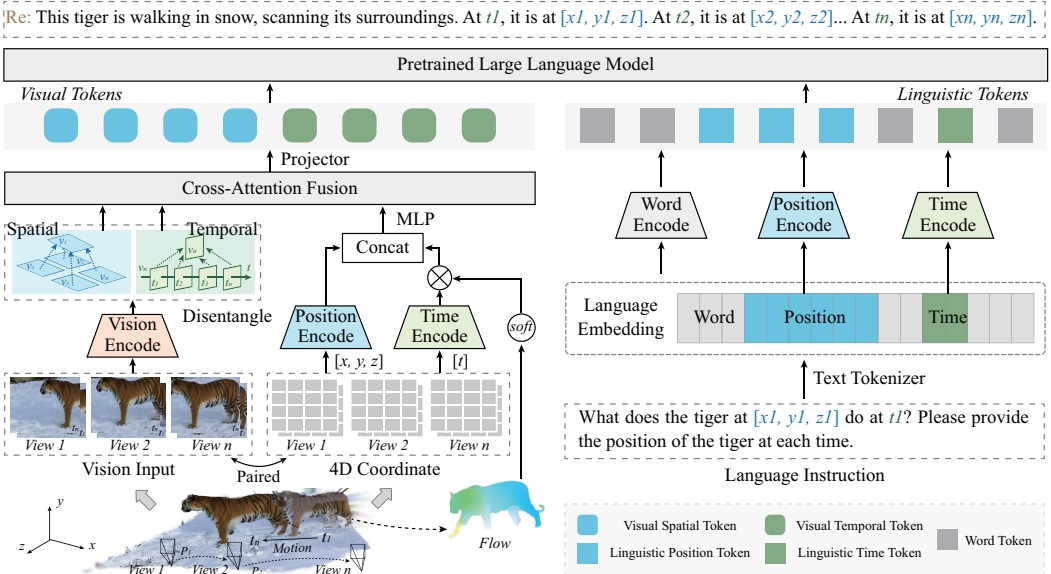

Figure 2: Our LLaVA-4D consists of three stages: 1) **4D coordinate encoding.** Encode 3D position and 1D time with optical flow. 2) **Vision embedding.** Disentangle visual features into spatiotemporal features and embed the encoded 4D coordinates via cross-attention fusion. 3) **Language embedding.** Align textual position and time with the fused vision embedding for 4D scene understanding.

3) **Coordinate-Aligned Language Embedding (*cf.* Sec. 3.3).** This is the linguistic representation stage where visual spatiotemporal features are projected into the language embedding space using multi-layer perception: $\tau_v^{st} = \text{MLP}(f_{st})$, where $\tau_v^{st}$ denotes visual spatiotemporal tokens. Input instructions are tokenized into language space, where textual position and time are spatiotemporally encoded into corresponding linguistic tokens $\tau_l^{st}$.

**Remarks.** The output $\tau_v^{st}$ and $\tau_l^{st}$ denote the visual and linguistic representation with 4D coordinate prompt, respectively. Subsequently, the LLM utilizes these enhanced visual and linguistic tokens to improve 4D scene understanding. Our unified framework embeds 4D coordinates as spatiotemporal prompts into visual representations to enable spatiotemporal understanding of dynamic objects and static background in LMMs. The following sections detail the design of each stage.

## 3.1 DYNAMIC-AWARE 4D COORDINATE ENCODING

2D LMMs can capture spatial relationship between targets in images by using a vision encoder to implicitly or explicitly incorporate 2D position encoding. Similarly, 4D scene understanding can be enabled by the integration of 3D positon and 1D time into LMMs.

**4D Coordinate Definition.** Given an image from a certain view at timestamp $t$, we use SfM (Schonberger & Frahm, 2016) for camera pose $P = [R \mid T]$ and MVS (Seitz et al., 2006) for depth $D$. Combined with intrinsic parameter $K$, we transform 2D pixel coordinate $x_{2D}$ to world coordinate system via geometric projection (Zou et al., 2018; Zhou et al., 2017):

$$x_{3D} = R^{-1}(D(x_{2D}) \cdot K^{-1}x_{2D} - T), \tag{4}$$

where $x_{3D} = [x, y, z]^\top$ denotes 3D position. After traversing all videos, we concatenate time and corresponding 3D position to form the 4D coordinate tensor $[x, y, z, t]$.

**Spatiotemporal Encoding.** We perform spatiotemporal encoding to convert the 4D coordinates into learnable feature patterns. It is challenging to directly distinguish objects from the background solely based on spatial dimensions such as multi-view images captured at a specific time. We circumvent this challenge by adopting the same spatial position encoding strategy for objects and background via learnable Fourier feature (Li et al., 2021):

$$p_{xyz} = \text{PE}(x, y, z) = 1/\sqrt{d} \, [\cos([x, y, z]W_r^\top \parallel \sin([x, y, z]W_r^\top))], \tag{5}$$

where $d$ denotes the dimension and $W_r$ is the learnable parameter of the Fourier feature. From the temporal dimension such as continuous video sequence at a certain view, objects and background

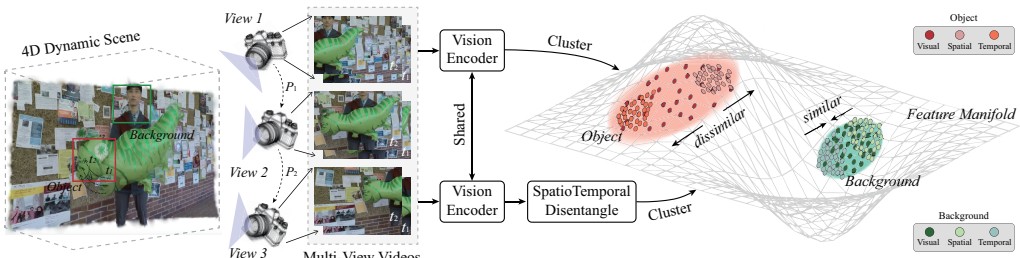

Figure 3: Feature distribution of static background and dynamic object in a 4D dynamic scene. Visual features of dynamic objects appear scattered while static backgrounds are clustered. In contrast, spatiotemporal features show clear discrimination between objects and background.

have different motion patterns and thus we add motion information into the temporal encoding:

$$p_t = \text{TE}(t) \cdot \beta = 1/\sqrt{d} \left[ \cos(tW_r^\top \parallel \sin(tW_r^\top)) \right] \cdot (1 + \Phi(\beta)), \qquad (6)$$

where $\beta$ is an estimated optical flow, and $\Phi(\cdot)$ is softmax function. Note that the optical flow is used as an auxiliary motion cue for temporal encoding instead of as the sole source of temporal information for dynamic scene understanding. We further concatenate the spatial and temporal encoding outputs to obtain dynamic-aware 4D coordinate embeddings as the spatiotemporal prompt for subsequent visual fusion. Moreover, this spatiotemporal prompt is extensible, where we can further add some other spatiotemporal attributes such as semantic and action to guide the alignment between visual and linguistic representations (*cf.* Sec. 5.2 for details).

### 3.2 SPATIOTEMPORAL-DISENTANGLED VISION EMBEDDING

Unlike 2D image and 3D scene, 4D scene consists of spatial such as color and temporal such as motion components. A unified visual representation for 4D scene usually suffers from misaligned heterogeneous features, which inspires us to disentangle visual features into spatiotemporal components.

**Spatiotemporal Disentanglement.** Spatial features mainly reflect the appearance of the entire scene, while temporal features focus more on continuous varying in motion patterns. After obtaining multi-view visual features $f_{v,t}$, where $v$ is the view and $t$ is the time, we get the correlation of visual features between different views at the same time as spatial features:

$$f_s = \text{Aggregate}(\{f_{v=i,t}^\top f_{v=j,t} \mid i \neq j\}). \qquad (7)$$

Next, we further calculate the correlation of visual features between adjacent images of continuous time at the same view as temporal features:

$$f_t = \text{Aggregate}(\{f_{v,t=i}^\top f_{v,t=i+1}\}). \qquad (8)$$

To illustrate the importance of spatiotemporal features, we encode the selected object and background regions of multi-view videos into visual, spatial and temporal features, and cluster these features for visualization in Fig. 3. The visual features of the object region appear scattered, but the spatiotemporal features of both the object and background regions are clustered. This indicates that spatiotemporal features are highly discriminative for the entire scene. Consequently, disentangling visual features into spatiotemporal components is essential for effective 4D scene representation.

**Cross-Attention Fusion.** Single disentangled spatiotemporal features cannot be localized to world coordinate system. We need to further embed 4D coordinates into the spatiotemporal features for localization. We first introduce an MLP to make the dimension of the 4D coordinate embeddings the same as the dimension of the spatiotemporal features: $p_{4D} = \text{MLP}([p_{xyz} \parallel p_t])$. Next, we fuse the 4D coordinate embeddings with the spatiotemporal features via a cross-attention mechanism:

$$
\begin{aligned}
q &= w_q p_{4D}, \quad k = w_k \, [f_s, f_t], \quad v = w_v \, [f_s, f_t], \quad a = \text{softmax}(qk^\top/\sqrt{d}), \\
o &= a \cdot v, \quad \alpha = \sigma(\text{MLP}_{obj}(p_{4D})), \quad f_{st} = \alpha \cdot o + (1-\alpha) \cdot f_s,
\end{aligned}
\qquad (9)
$$

where $w$ is a learnable weight. As a result, we can obtain the 4D-aware visual spatiotemporal feature.

### 3.3 COORDINATE-ALIGNED LANGUAGE EMBEDDING

Although vision embeddings can represent scene knowledge, LMMs understanding scenes require the alignment between visual and linguistic representations. Since the input of large language model

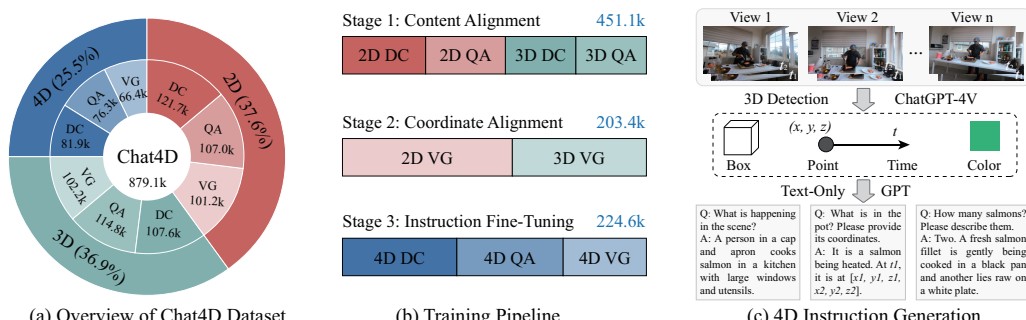

(a) Overview of Chat4D Dataset     (b) Training Pipeline     (c) 4D Instruction Generation

Figure 4: Overview of our dataset and training pipeline. (a) Chat4D dataset includes 2D, 3D, and 4D vision-language training sets for dense captioning, QA, and visual grounding. (b) Three-stage training: stages 1-2 use 2D/3D data for initialization; stage 3 uses 4D data for instruction fine-tuning. (c) Spatiotemporal characteristics are extracted as local descriptions to generate 4D instructions.

requires text-like tokens, we first build a multi-layer perception to project the fused spatiotemporal features into language embedding space for preliminary alignment with linguistic representations. This projected fused spatiotemporal features is the visual tokens denoted as $\tau_v^{st}$. Subsequently, we tokenize the input instruction into the language space with word tokens $\tau_l$, and apply the same spatiotemporal encodings $\text{PE}(\cdot)$ and $\text{TE}(\cdot)$ to textual position $tp$ and time $tt$:

$$\tau_s = \text{PE}(tp), \tau_t = \text{TE}(tt). \tag{10}$$

We further fuse the encoded position and time with corresponding word token: $\tau_l^{st} = \tau_l + w_s\tau_s + w_t\tau_t$, where $w_s$ and $w_t$ denote the learnable weights. We concatenate 4D-aware visual tokens with coordinate-aligned linguistic tokens for the LLM to reason. Particularly, 4D coordinate encoding ensures spatiotemporal localization of the scene within our unified framework. Additionally, disentangled vision embedding models spatiotemporal knowledge of the scene, and vision-language alignment further enable the LMM to achieve interactive understanding of 4D scenes.

## 4 DATASET AND TRAINING PIPELINE

### 4.1 OUR CHAT4D DATASET

Many vision-language datasets have been proposed to evaluate 2D/3D scene understanding of LMMs. However, there is currently no vision-language dataset specifically designed for 4D scene understanding in LMMs. To address this gap, we introduce the Chat4D dataset. As illustrated in Fig. 4(a-b), our dataset includes 2D, 3D and 4D vision-language data types, where 2D/3D data are used to initialize multimodal spatiotemporal understanding and 4D data is used for instruction fine-tuning.

**2D&3D Vision-Language Data.** To develop multimodal spatiotemporal understanding, our model requires image-format inputs and a large number of vision-language pairs. We thus integrate existing standard 2D/3D spatiotemporal datasets (Chen et al., 2024b; Wang et al., 2023a; Luo et al., 2023; Azuma et al., 2022; Lyu et al., 2024; Chen et al., 2021; Zhang et al., 2023) and adapt specific text instructions (*e.g.*, 2D/3D position and time) to align with our spatiotemporal encoding strategy. This approach effectively trains the LMM for spatial and temporal understanding. These datasets cover dense captioning (DC), visual QA and visual grounding (VG) tasks with a total of 654.5K samples.

**4D Vision-Language Data.** We merge existing 4D dynamic scene reconstruction datasets to train the LMM for 4D spatiotemporal understanding: iPhone (Gao et al., 2022), HyperNeRF (Park et al., 2021), N3DV (Li et al., 2022b), PanopticSports (Luiten et al., 2024), DAVIS (Perazzi et al., 2016), and Immersive (Broxton et al., 2020)). Note that most videos in these datasets last 6-12 seconds. Additionally, we develop a data generation approach to produce paired 4D vision-language data for instruction fine-tuning. As shown in Fig. 4(c), we utilize the 3D object detection method (Rukhovich et al., 2022) and GPT-4V (Yang et al., 2023) to extract local spatiotemporal information such as category, position, time from multi-view videos. These extracted features are then processed by text-only GPT to generate global 4D descriptions in instruction-following formats for typical understanding tasks to produce a dataset of 224.6K samples. To further improve label quality, we apply two rounds of data cleaning, such as automatic filtering and manual inspection. For the automatic filtering, we enforce temporal consistency and spatial overlap constraints to remove 4.7%

Table 1: Quantitative results of LMMs for scene understanding tasks on different 3D and 4D datasets.

| | Methods | 3D Benchmark | | | | | | | | 4D Benchmark | | | |
| | | Scan2Cap | | | ScanQA | | | Multi3DRefer | ScanRef | Chat4D (Ours) | | | |
| | | C@0.5↑ | B-4@0.5↑ | M@0.5↑ | C↑ | B-4↑ | M↑ | F1@0.5↑ | SAcc@0.5↑ | C↑ | B-4↑ | SAcc@0.5↑ | TAcc↑ |
|---|---|---|---|---|---|---|---|---|---|---|---|---|---|
| 3D | 3D-LLM | – | – | – | 69.4 | 12.0 | 14.5 | – | – | 61.6 | 11.5 | 31.4 | – |
| | Chat-3D v2 | 63.9 | 31.8 | – | 87.6 | 14.0 | – | 41.6 | 38.4 | 81.8 | 13.7 | 39.5 | – |
| | LL3DA | 65.2 | 36.8 | 26.0 | 76.8 | 13.5 | 15.9 | – | – | 72.3 | 11.9 | 46.2 | – |
| | 3D-LLaVA | 78.8 | 36.9 | 27.1 | 92.6 | 17.1 | 18.4 | – | – | 85.1 | 16.0 | 52.0 | – |
| | Grounded 3D-LLM | 70.6 | 35.5 | – | 72.7 | 13.4 | – | 40.6 | 44.1 | 66.3 | 12.2 | 43.7 | – |
| | PQ3D | 80.3 | 36.0 | 29.1 | 87.8 | – | 17.8 | 50.1 | 51.2 | 84.7 | 14.3 | 51.5 | – |
| | LLaVA-3D | 79.2 | 41.1 | 30.2 | 91.7 | 14.5 | 20.7 | – | 42.2 | 87.4 | 14.8 | 45.6 | – |
| | Video-3D LLM | 83.8 | 42.4 | 28.9 | **102.1** | 16.2 | 19.8 | 52.7 | 51.7 | 89.4 | 16.1 | 52.8 | – |
| | Spatial-MLLM | – | – | – | 91.8 | 14.8 | 18.4 | – | – | – | – | – | – |
| | 3UR-LLM | – | – | – | 87.7 | 15.5 | 18.4 | – | – | – | – | – | – |
| | GPT-4o w/ Co. Corr. | – | – | – | 87.0 | – | 18.0 | – | – | – | – | – | – |
| 4D | LLaVA-4D (Ours) | **85.3** | **45.7** | **31.3** | 97.8 | **17.9** | **21.2** | **54.3** | **53.2** | **93.5** | **17.2** | **58.9** | **54.6** |

Table 2: Quantitative results of LMMs for scene understanding tasks on VSI-Bench.

| Methods | Average | Numerical Answer | | | | Multiple-Choice Answer | | | |
| | | Obj. Count | Abs. Dist. | Obj. Size | Room Size | Rel. Dist. | Rel. Dir. | Route Plan | Appr. Order |
|---|---|---|---|---|---|---|---|---|---|
| LLaVA-Video-7B | 35.6 | 48.5 | 14.0 | 47.8 | 24.2 | 43.5 | 42.4 | **34.0** | 30.6 |
| LLaVA-OneVision-7B | 32.4 | 47.7 | 20.2 | 47.4 | 12.3 | 42.5 | 35.2 | 29.4 | 24.4 |
| LongVA-7B | 29.2 | 38.0 | 16.6 | 38.9 | 22.2 | 33.1 | 43.3 | 25.4 | 15.7 |
| VILA-1.5-8B | 28.9 | 17.4 | 21.8 | 50.3 | 18.8 | 32.1 | 34.8 | 31.0 | 24.8 |
| InternVL2-8B | 37.5 | 31.3 | 29.0 | 48.9 | 44.2 | 38.0 | 33.4 | 28.9 | 46.4 |
| Spatial-MLLM | 48.4 | 65.3 | 34.8 | 63.1 | 45.1 | 41.3 | **46.2** | 33.5 | 46.3 |
| LLaVA-4D (Ours) | **48.6** | **68.2** | **35.3** | **64.8** | **49.6** | **44.5** | 45.2 | 33.8 | **47.5** |

of the samples with inconsistent motion, impossible object trajectories, or mismatched spatial regions. For the manual inspection, we manually discard 0.8% of the remaining samples with abnormal labels, including timestamp misalignment, large coordinate deviations, and incorrect color or attribute descriptions. The two steps substantially increase the reliability of the final annotations.

## 4.2 TRAINING PIPELINE

To ensure the stability of the training process and improve the performance of the model, we divide the entire training into three stages in Fig. 4 (b) as follows:

**Stage 1: Content Alignment.** The training sets of the DC and QA tasks in the 2D&3D vision-language data of our Chat4D are used to initially align the content between visual and linguistic representations. This provides a foundational spatiotemporal understanding for the proposed model. At this stage, only parameters of the cross-attention fusion and the projector are updated. The 4D spatiotemporal coordinate features $p_{4D}$ are temporarily set as zero padding.

**Stage 2: Spatiotemporal Coordinate Alignment.** In order to further improve the fine-grained understanding capability of our model under the spatiotemporal coordinate prompt, we use the training data of the VG task in the 2D&3D vision-language subset of our Chat4D to refine the spatiotemporal coordinate alignment between visual and linguistic representations. At this stage, we update all trainable parameters of 4D coordinate encoding and cross-attention fusion modules while keeping all other modules frozen.

**Stage 3: 4D Task Instruction Fine-Tuning.** To further improve our model for 4D scene understanding, we use 4D vision-language data of Chat4D to enhance the generalization of our model for fine-grained spatiotemporal understanding with 4D coordinates through a multi-task instruction fine-tuning strategy. All trainable parameters are updated while the vision encoder remains frozen.

## 5 EXPERIMENTS

**Implements Details.** Our LLaVA-4D utilizes the pre-trained weights of LLaVA-1.5-7B (Liu et al., 2024) and the vision encoder of CLIP-ViT-L-336px (Radford et al., 2021). Cross-attention fusion module is a transformer-based architecture. The whole model is trained on 8 RTX 4090 GPUs over 86 hours using AdamW as the optimizer. In training stages 1-2, we set the learning rate to $1.0e - 4$ with a batch size of 48. We use a learning rate of $1.0e - 5$ with a batch size of 16 in training stage 3.

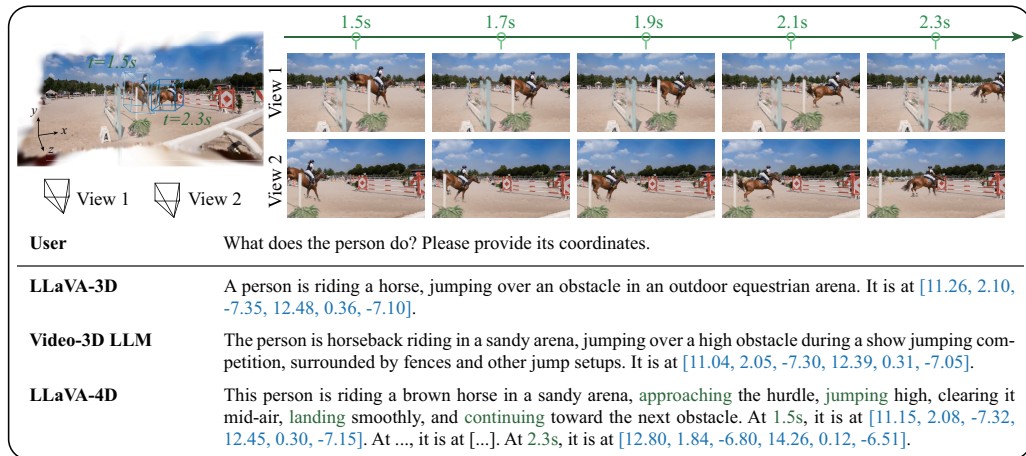

Figure 5: Visual comparison of LMMs on 4D scene understanding.

**Comparison Methods.** Since there are currently no public 4D LMMs for comparison, we compare our model with 3D LMMs: 3D-LLM (Hong et al., 2023), Chat-3D v2 (Huang et al., 2023), LL3DA (Chen et al., 2024a), 3D-LLaVA (Deng et al., 2025), Grounded 3D-LLM (Chen et al., 2024c), PQ3D (Zhu et al., 2024b), LLaVA-3D (Zhu et al., 2024a), Video-3D LLM (Zheng et al., 2024), Spatial-MLLM (Wu et al., 2025), 3UR-LLM (Xiong et al., 2025), and GPT-4o with Coarse Correspondences (Liu et al., 2025). For a fair comparison, all methods are fine-tuned on the same evaluation benchmark.

**Evaluation Metric.** We compare all competing methods on multiple 3D datasets: Scan2Cap (Chen et al., 2021), ScanQA (Azuma et al., 2022), ScanRef (Chen et al., 2020) and Multi3DRefer (Zhang et al., 2023) and our Chat4D dataset. We evaluate the quality of generated text response for Scan2Cap and ScanQA in terms of CiDEr (C), BLEU-4 (B-4), METEOR (M). We choose the F1 metric of object prediction precision for Multi3DRefer, and the accuracy of intersection over unions for grounding task from ScanRef. The metrics are also applicable to the evaluation on our Chat4D, where grounding accuracy is divided into spatial and temporal components: S/TAcc.

### 5.1 COMPARISON WITH STATE-OF-THE-ART MODELS

**Quantitative Results.** In Table 1, we compare the competing methods on 3D and 4D datasets. For 3D understanding comparison, our method performs better than other methods. This shows that spatial features disentangled from multi-view images have stronger representation than ordinary visual features for 3D scene. For 4D understanding comparison, our method achieve a significant strength due to dynamic-aware 4D coordinate as spatiotemporal prompt.

**Comparison on VSI-Bench.** In Table 2, we introduce VSI-Bench (Yang et al., 2025) and compare our model with several representative LLMs: LLaVA-Video-7B (Zhang et al., 2024c), LLaVA-OneVision-7B (Li et al., 2024), LongVA-7B (Zhang et al., 2024b), VILA-1.5-8B (Lin et al., 2024), InternVL2-8B (Chen et al., 2024e), and LongVILA-8B (Chen et al., 2024d). The results show that our LLaVA-4D achieves the best performance on most metrics, with only a few metrics slightly lower than competing methods. This verifies the superiority of our model in multimodal spatial reasoning.

**Qualitative Results.** In Fig. 5, we select a typical 4D scene from our Chat4D dataset to visualize the comparison between our model and 3D LMMs. The results show that 3D LMMs cannot respond to timestamp and corresponding temporal information, while our method can understand the temporal content. This is because 3D LMMs lack the representation of temporal characteristic. In contrast, our method introduces the spatiotemporal prompt to enhance the dynamic representation of 4D scenes.

**Comparison on Temporal Understanding.** In Table 3, we select a subset of video clips containing typical actions and obtain the corresponding time intervals for these actions to compare our model with several video-based LMMs: LLaVA-ST (Li et al., 2025a) and Grounded-VideoLLM (Wang et al., 2024). The results show that our tIoU scores are comparable to those of the baseline models, but our spatiotemporal understanding metrics,

Table 3: Comparison on temporal understanding.

| Method | SAcc@0.5↑ | TAcc↑ | tIoU@0.5↑ |
|---|---|---|---|
| Grounded-VideoLLM | 9.4 | 5.1 | 47.0 |
| LLaVA-ST | 15.2 | 7.3 | 58.7 |
| LLaVA-4D (Ours) | **58.9** | **54.6** | **61.5** |

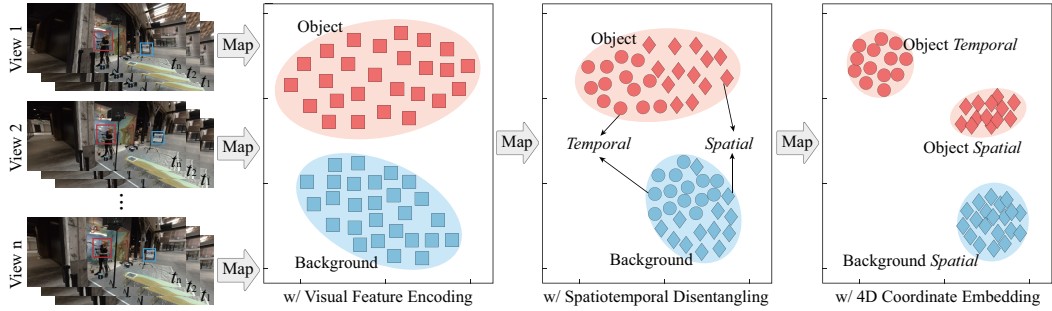

Figure 6: Feature visualization at different stages. Spatiotemporal disentanglement improves the discriminability of background and objects, which are further separated by 4D coordinate embedding.

such as SAcc and TAcc, are significantly higher. This is because tIoU mainly depends on video-level timestamp annotations, whereas SAcc and TAcc rely on 4D spatiotemporal coordinate understanding. Therefore, these results further demonstrate that our LLaVA-4D represents a promising paradigm with fine-grained multimodal spatiotemporal reasoning capabilities for 4D dynamic scene understanding.

## 5.2 ABLATION STUDY AND DISCUSSION

**Effect of Visual Representation Modules.** In Table 4, we verify the effectiveness of coordinate embedding, feature disentanglement and feature fusion modules in visual representation. Coordinate embedding is the key to improving the overall performance of 4D understanding

Table 4: Effect of visual representation modules.

| Coor. embed | Feat. disent. | Feat. fusion | C↑ | B-4↑ | SAcc@0.5↑ | TAcc↑ |
|---|---|---|---|---|---|---|
| × | × | × | 62.3 | 11.7 | 34.8 | 12.7 |
| ✓ | × | × | 85.4 | 15.1 | 51.5 | 47.5 |
| ✓ | ✓ | × | 89.0 | 16.5 | 54.3 | 51.2 |
| ✓ | ✓ | ✓ | **93.5** | **17.2** | **58.9** | **54.6** |

tasks by a large margin. Feature disentanglement improves the upper limit of 4D scene understanding to a certain extent by strengthening the representation of spatial and temporal characteristics. Feature fusion further enhances the spatiotemporal understanding ability of the LMM.

**Role of 4D Coordinate Encoding.** In Table 5, we analyze the impact of 3D position encoding and 1D time encoding on the performance of 4D understanding. When 4D coordinates are not encoded, the spatiotemporal understanding performance of the LMM is negatively affected

Table 5: Role of coordinate encoding.

| Encoding target | C↑ | B-4↑ | SAcc@0.5↑ | TAcc↑ |
|---|---|---|---|---|
| w/o Encoding | 75.0 | 12.1 | 47.2 | 46.8 |
| w/ 3D position | 88.6 | 15.3 | 53.4 | 47.0 |
| w/ 1D time | 82.7 | 14.0 | 48.5 | 52.7 |
| w/ 4D coordinate | **93.5** | **17.2** | **58.9** | **54.6** |

to a certain extent. 3D position encoding mainly contributes to the spatial understanding ability of the LMM, and 1D time encoding can further improve the performance of temporal understanding.

**How Spatiotemporal Features Work?** We study how spatiotemporal features work within our model in Fig. 6. Initially, the visual features of objects and backgrounds appear relatively scattered. After spatiotemporal disentanglement, object features are distinctly divided into spatial and temporal components with background features remain clustered. Incorporating coordinate embedding further organizes object features into two distinct sets: spatial and temporal where background features consolidate into a unified spatial set. This demonstrates the strong spatiotemporal representation capability of our method for dynamic objects and static backgrounds.

**Choice of Spatiotemporal Fusion Strategy.** Table 6 shows attention-based fusion outperforms concatenation and weighting fusion strategies. This is because concatenation and weighting rely on global unified fusion with fixed

Table 6: Discussion on spatiotemporal fusion.

| Fusion strategy | C↑ | B-4↑ | SAcc@0.5↑ | TAcc↑ |
|---|---|---|---|---|
| w/ Concatenation | 89.0 | 16.5 | 54.3 | 51.2 |
| w/ Weighting | 89.5 | 16.5 | 55.1 | 51.4 |
| w/ Attention | **93.5** | **17.2** | **58.9** | **54.6** |

weights. In contrast, attention-based fusion can dynamically adjust the fusion weights of spatiotemporal features according to 4D coordinate embedding. This allows the LMM to effectively focus more on meaningful spatiotemporal features for 4D understanding.

**Extensibility of Spatiotemporal Prompt.** To verify that the spatiotemporal prompt in our model is an extensible feature, we introduce additional spatial semantic and temporal action masks as prompts based on the encoded 4D coordinates to train our model. As shown in Fig. 7, our model can reason

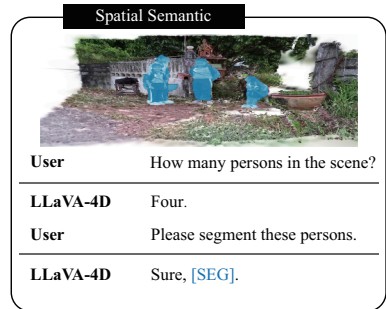
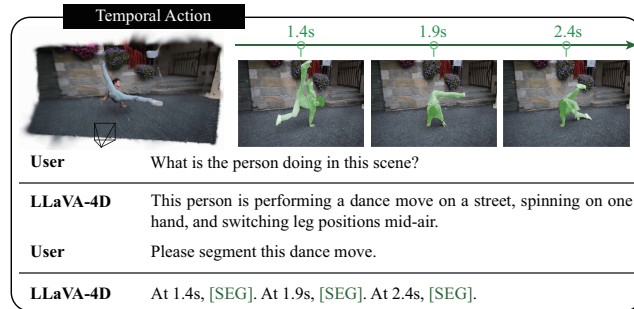

Figure 7: Visualization of spatiotemporal prompt extended to other spatiotemporal vision tasks.

about the visual features of semantic and action based on specific text instructions. Incorporating spatiotemporal prompts thus enhances the generality of our model for various vision tasks.

**Impact of Textual Coordinate Encoding.** Table 7 ablates the impact of textual coordinate encoding on scene understanding by the LMM. We deduce: 1) Textual coordinates as instructions improve the fine-grained spatiotemporal understanding of the LMM. 2) Textual coordinate encoding further improves the upper limit of 4D spatiotemporal understanding. This is because coordinate encoding helps minimize the risk of LLM misinterpreting coordinate values.

Table 7: Impact of textual coordinate encoding.

| Text instruction | | C↑ | B-4↑ | SAcc@0.5↑ | TAcc↑ |
|---|---|---|---|---|---|
| w/o Coordinate | | 83.5 | 13.2 | 43.2 | 25.8 |
| w/ Coordinate | w/o Encoding | 90.1 | 16.7 | 56.3 | 53.0 |
| | w/ Encoding | **93.5** | **17.2** | **58.9** | **54.6** |

**Robustness to Annotation Errors.** In Table 8, we discuss the impact of annotation errors, where we inject small random perturbations into the 3D coordinates and timestamps using mild Gaussian noise, and fine-tune our model on the perturbed version of Chat4D. The results show that the drops across all metrics are within 1 point, which is small relative to the absolute performance level. This suggests that our LLaVA-4D is robust to minor spatiotemporal noise in the annotations.

Table 8: Discussion on impact of annotation errors.

| Fine-tuning strategy | C↑ | B-4↑ | SAcc@0.5↑ | TAcc↑ |
|---|---|---|---|---|
| Our base LLaVA-4D | 93.5 | 17.2 | 58.9 | 54.6 |
| + Fine-tuned on perturbed Chat4D | 93.0 (0.5↓) | 17.0 (0.2↓) | 58.1 (0.8↓) | 53.9 (0.7↓) |

**Effect of Temporal Encoding Strategy.** In Table 9, we analyze the effect of different temporal encoding strategies. We can observe that frame rate-based encoding performs worse than the motion speed-based encoding, especially in fine-grained temporal understanding. The underlying reason is that the frame rate-based approach uses a globally fixed-step encoding, which struggles to explicitly capture the temporal characteristics of independently moving objects in dynamic scenes. In contrast, our motion speed-based approach is a locally adaptive encoding scheme that is capable of directly representing the magnitude of motion, and thus reflecting the true physical dynamics of the scene.

Table 9: Effect of temporal encoding strategy.

| Temporal encoding strategy | C↑ | B-4↑ | SAcc@0.5↑ | TAcc↑ |
|---|---|---|---|---|
| Frame rate-based encoding | 91.0 | 16.7 | 57.3 | 49.5 |
| Motion speed-based encoding | **93.5** | **17.2** | **58.9** | **54.6** |

**Limitation.** Our model has achieved success in end-to-end inference for scene understanding, but the preprocessing stage related to 4D coordinate construction relies on external modules, *e.g.* SfM. This slightly reduces the overall efficiency. Since the preprocessing stage is replaceable, we plan to adopt end-to-end geometric models, *e.g.* MonST3R (Zhang et al., 2024a) to improve practicality.

## 6 CONCLUSION

In this work, we propose LLaVA-4D, a first general vision-language LMM for 4D scene understanding. We introduce a dynamic-aware 4D coordinate encoding as a spatiotemporal prompt for scene content localization. Additionally, we propose a spatiotemporal-disentangled vision embedding method that integrates 4D spatiotemporal prompts into disentangled spatiotemporal features for effective scene representation. By aligning visual spatiotemporal embeddings with language embeddings, our approach allows LMMs to comprehend the 4D physical world. To support training, we construct Chat4D, a comprehensive dataset covering 2D, 3D and 4D vision-language data for multimodal spatiotemporal understanding. Extensive experiments validate the effectiveness of our method.

## ACKNOWLEDGMENTS

This research work is supported by the Tier 2 grant MOE-T2EP20124-0015 from the Singapore Ministry of Education.

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
