# OpenReview forum: "LLaVA-4D: Embedding SpatioTemporal Prompt into LMMs for 4D Scene Understanding"
_ICLR.cc/2026/Conference — ICLR 2026 Poster_

### Official Review · Reviewer_NHKE · 2025-10-31

**Soundness:** 3
**Presentation:** 3
**Contribution:** 3
**Rating:** 6
**Confidence:** 3

**Summary:**

In this paper, the authors introduce LLaVA-4D, an advanced vision-language large model designed specifically for comprehensive 4D scene understanding. By leveraging the robust LLaVA multimodal framework, the authors’ primary innovation involves the incorporation of a novel spatiotemporal prompt. This innovation blends static 3D spatial perception with dynamic temporal dimension information, enhancing the model's capability to grasp scene dynamics. Specifically, LLaVA-4D encodes both 3D spatial coordinates and time into cohesive 4D coordinate embeddings, subsequently embedding these into visual features that have been temporally and spatially decoupled. This integration aligns these refined visual features with corresponding language embeddings, significantly augmenting the model's comprehension of dynamic scenes. Furthermore, to facilitate training and evaluation, the authors developed Chat4D, a comprehensive 4D vision-language dataset featuring detailed spatiotemporal coordinate annotations. Experimental findings underscore LLaVA-4D's superiority over existing 3D language-multimodal models (LMMs) in tackling 3D tasks and its clear leadership in 4D tasks, marking a significant advancement in 4D scene understanding.

**Strengths:**

1.	The LLaVA-4D model is a groundbreaking innovation, endowing large language models with 4D understanding, transcending previous 3D-focused limitations.
2.	The authors have a clear reason for decoupling spatiotemporal visual features, and ablation experiments also prove that this decoupling module can make the representation of spatiotemporal features stronger.
3.	The Chat4D dataset, developed by the authors, addresses a critical void in 4D scene understanding for multimodal large language models. This contribution promises to enhance future research endeavors positively.

**Weaknesses:**

1.	The paper lacks clarity on LLaVA-4D's capabilities, failing to specify its maximum video frame rate and duration. Additionally, it omits critical details about the 4D data in the Chat4D dataset, such as the average video length.
2.	When comparing with state-of-the-art models, it seems that other models haven’t been trained or fine-tuned on Chat4D. (As shown in Table 1, they don’t even have the ability to output information related to the temporal dimension.) This makes it hard to directly show that LLaVA-4D is better than other models at understanding the temporal dimension.
3.	When testing the model’s ability to understand the temporal dimension, the study only uses TAcc as the evaluation metric. We need more evaluation methods—like action interval prediction and the corresponding measurement metrics, for example.

**Questions:**

1.	The 4D benchmark in Chat4D is mainly based on 4D dynamic scene reconstruction datasets. The authors used GPT-4V and text-only GPT to generate a lot of global 4D descriptions. During this process, how did the authors assess and fix the hallucination problem of large models?
2.	Will the code and dataset be made open-source later?

---

> ### Author Response · Authors · 2025-11-20
> **Strength Summary and Rebuttal to Q1, Q2**
>
> Thanks for recommending our strengths: **groundbreaking and innovative** model that **transcending** previous 3D-focused limitations, **stronger** disentangled visual features with a **clear reason** and ablation experiments, **contributed** Chat4D dataset that addresses a **critical void** in 4D understanding and **advances future research**.
>
> Regarding the raised weaknesses and questions, we summarize them into four aspects: model and dataset details, fairness of comparison, additional temporal evaluation and hallucination problem. We address these weaknesses in detail.
>
>
>
> **Q1:** **[Model and Dataset Details]** The paper lacks clarity on LLaVA-4D's capabilities, failing to specify its maximum video frame rate and duration. Additionally, it omits critical details about the 4D data in the Chat4D dataset, such as the average video length.
>
> **A:** The specific details of the proposed model and dataset are described as follows:
>
> - Regarding the video frame rate, we have discussed the impact of different frame rates on model performance and efficiency in Table 9 of the supplementary material to demonstrate the threshold effect of frame rate on model performance. Specifically, a higher frame rate does not always lead to better performance. Beyond 30 FPS, the performance gain saturates while storage demand increases sharply. Therefore, a reasonable frame rate can be set within the range of 15–30 FPS.
>
> - Regarding the maximum duration, our model currently handles videos of up to 14 seconds. Longer durations introduce long-term causal reasoning challenges that can lead to temporal forgetting, significantly degrading the spatiotemporal understanding capability of our model.
>
> - Regarding the video length in the dataset, most videos last 6-12 seconds. Although some sequences in the original 4D reconstruction datasets are longer, we perform frame sampling or slicing to ensure that each video segment used for model fine-tuning remains within 14 seconds.
>
>
>
> **Q2:** **[Comparison Fairness]** When comparing with state-of-the-art models, it seems that other models haven’t been trained or fine-tuned on Chat4D. (As shown in Table 1, they don’t even have the ability to output information related to the temporal dimension.) This makes it hard to directly show that LLaVA-4D is better than other models at understanding the temporal dimension.
>
> **A:** Thanks for your careful review. For the training of the competing methods, we have briefly mentioned in Line 399 of the main text that all competing models are fine-tuned on the same evaluation benchmark. Considering that the competing 3D LMMs mainly take single-view or multi-view images of static scenes as input, they lack temporal encoding and dynamic information embedding. If these 3D LMMs are trained directly using text descriptions containing temporal information, this would introduce spurious reasoning cues and even weaken their original scene understanding capability. Therefore, during training of the comparison methods, we fine-tune the 3D LMMs using the complete Chat4D dataset, but with temporal descriptions masked out. Under this training setup, the quantitative results in Table 1 of the main text can fairly and effectively demonstrate that our LLaVA-4D possesses an accurate temporal understanding ability that 3D LMMs inherently lack.

---

> ### Author Response · Authors · 2025-11-20
> **Rebuttal to Q3, Q4 and Q5**
>
> **Q3:** **[Temporal Evaluation]** When testing the model’s ability to understand the temporal dimension, the study only uses TAcc as the evaluation metric. We need more evaluation methods—like action interval prediction and the corresponding measurement metrics, for example.
>
> **A:** Thanks for your valuable suggestion. In our proposed Chat4D dataset, we select a subset of video clips containing typical actions and obtain the corresponding time intervals for these actions. Based on these intervals, we generate text descriptions related to action interval prediction to fine-tune our model. We then compare our model with several video-based LMMs (*e.g.*, LLaVA-ST [1], Grounded-VideoLLM [2]) and measure the quantitative metric tIoU for evaluating action interval prediction, as shown in the following table:
>
> |      Method       | SAcc$\uparrow$ | TAcc$\uparrow$ | tIoU@0.5$\uparrow$ |
> | :---------------: | :------------: | :------------: | :----------------: |
> | Grounded-VideoLLM |      9.4       |      5.1       |        47.0        |
> |     LLaVA-ST      |      15.2      |      7.3       |        58.7        |
> |  LLaVA-4D (Ours)  |    **58.9**    |    **54.6**    |      **61.5**      |
>
> The results show that our tIoU scores are comparable to those of the baseline models, but our spatiotemporal understanding metrics, such as SAcc and TAcc, are significantly higher. This is because tIoU mainly depends on video-level timestamp annotations, whereas SAcc and TAcc rely on 4D spatiotemporal coordinate understanding. Therefore, these results further demonstrate that LLaVA-4D represents a promising paradigm with fine-grained multimodal spatiotemporal reasoning capabilities for 4D dynamic scene understanding.
>
>
>
> **Q4:** **[Hallucination Problem]** The 4D benchmark in Chat4D is mainly based on 4D dynamic scene reconstruction datasets. The authors used GPT-4V and text-only GPT to generate a lot of global 4D descriptions. During this process, how did the authors assess and fix the hallucination problem of large models?
>
> **A:** We acknowledge that hallucination is a common and inherent challenge for large multimodal models. In the current version of our work, we address this issue from two main perspectives: data filtering and prediction correction.
>
> **1) For the data filtering**, when generating 4D descriptions, we employ a multi-stage filtering and correction mechanism. On one hand, we use GPT-4V and other SOTA large multimodal models for visual verification, which ensures that the generated descriptions are consistent with the visual content. On the other hand, we perform spatiotemporal consistency checking to automatically remove text descriptions with temporal or spatial inconsistencies. Finally, we conduct random manual verification of selected 4D scene clips and their corresponding text descriptions. These steps collectively ensure the high quality of the Chat4D dataset and help mitigate hallucination during model training.
>
> **2) For the prediction correction**, we further perform consistency evaluation between the generated outputs from our model and the predictions from GPT-4V. Text descriptions showing large discrepancies are manually corrected, and the feedback is then used for further fine-tuning, effectively reducing hallucination in the model.
>
> In the future, we plan to investigate some techniques designed for network architecture to further strengthen hallucination mitigation.
>
>
>
> **Q5:** **[Code Release]** Will the code and dataset be made open-source later?
>
> **A:** As stated in the main text, we will release our dataset and code once the paper is accepted.

---

> ### Author Response · Authors · 2025-11-20
> **Reference**
>
> **Reference**
>
> [1] Li H, Chen J, Wei Z, et al. Llava-st: A multimodal large language model for fine-grained spatial-temporal understanding. Proceedings of the Computer Vision and Pattern Recognition Conference. 2025: 8592-8603.
>
> [2] Wang H, Xu Z, Cheng Y, et al. Grounded-videollm: Sharpening fine-grained temporal grounding in video large language models. arXiv preprint arXiv:2410.03290, 2024.

---

### Official Review · Reviewer_n3mW · 2025-11-01

**Soundness:** 3
**Presentation:** 3
**Contribution:** 3
**Rating:** 6
**Confidence:** 3

**Summary:**

This paper proposes a novel vision-language multimodal model that supports dynamic scenes. By embedding 4D spatiotemporal cues, LMMs can simultaneously understand the spatial features of static backgrounds and the temporal features of dynamic objects. A 4D spatiotemporal cue fused with optical flow is designed, and a 4D vision-language dataset containing 879.1K samples is constructed. Based on this dataset, a three-stage training pipeline is designed to enhance the model's understanding capabilities.

**Strengths:**

- The dataset makes a significant contribution, and the method can significantly improve the ability to understand 4D scenes.

- The writting is clear and motivation is easy to understand.

- The three-stage training avoids the convergence difficulties caused by directly training 4D features, enabling the model to smoothly transition from basic 2D/3D capabilities to 4D capabilities.

**Weaknesses:**

- Data quality: 4D data relies on GPT-4V to extract spatiotemporal information and GPT to generate instructions, which may contain annotation errors (such as coordinate deviation and timestamp misalignment), affecting the model fine-tuning effect. However, the paper does not evaluate the impact of annotation errors on performance.

- Temporal coding: it relies solely on optical flow to estimate motion information. However, optical flow is prone to inaccurate estimation in fast-moving or occluded scenes, which may lead to deviations in the temporal characteristics of dynamic objects and affect the accuracy of the model's trajectory prediction for high-speed moving objects.

- Lack of experiments: the contribution in TAcc of optical flow and the kind of temporal coding like frame rate-based vs. motion speed-based are unclear.

**Questions:**

- What is the inital of w_p and beta?

- The paper uses CLIP-ViT-L-336px as the visual encoder, but why not use a more suitable encoder for video (such as the Video Swin Transformer) was not chosen?

- What are the advantages of LMMs frameworks over pure vision models in 4D tasks, and how do they compare to 4D Gaussians and VG4D?

---

> ### Author Response · Authors · 2025-11-20
> **Strength Summary and Rebuttal to Q1**
>
> Thanks for appreciating our work: the proposed dataset with **significant contribution**, the proposed method with **significant improvement** on 4D scene understanding, **clear writing**, **easily-understood** motivation, three-stage training strategy that enables smooth 2D-4D transition.
>
> Regarding the comments raised by the reviewer, we summarize them into four aspects: impact of annotation errors, effect of temporal encoding, technical details and advantages of LMMs on 4D tasks. Next, we respond to these comments point by point.
>
>
>
> **Q1:** **[Impact of Annotation Errors]** Data quality: 4D data relies on GPT-4V to extract spatiotemporal information and GPT to generate instructions, which may contain annotation errors (such as coordinate deviation and timestamp misalignment), affecting the model fine-tuning effect. However, the paper does not evaluate the impact of annotation errors on performance.
>
> **A:** Thank you for raising this important concern about data quality. We address the impact of annotation errors from two angles: ***how we construct and clean the labels***, and ***how sensitive the model is to simulated noise***.
>
> **1) Label Construction and Data Cleaning**
>
> - **Spatial annotations.** 3D positions and bounding boxes come from a dedicated 3D object detection model [1]. These outputs provide metrically grounded spatial information, independent of GPT-4V.
> - **Temporal annotations.** Timestamps come from frame-level processing of videos. We extract temporal indices directly from the video stream rather than from GPT-based descriptions.
> - **Textual/color attributes.** GPT-4V only provides semantic attributes such as color descriptions.
>
> To further improve label quality, we apply ***two rounds of data cleaning***:
>
> - **Automatic filtering.** We enforce temporal consistency and spatial overlap constraints. This step removes samples with inconsistent motion, impossible object trajectories, or mismatched spatial regions.
> - **Manual inspection.** We manually discard residual abnormal labels, including: Timestamp misalignments, large coordinate deviations, and incorrect color or attribute descriptions.
>
> These steps target exactly the types of errors mentioned in the question (coordinate shifts and misaligned timestamps) and substantially increase the reliability of the final annotations.
>
> **2) Experimental sensitivity to annotation noise**
>
> To explicitly measure the impact of annotation errors, we ***simulate*** plausible noise patterns and retrain the model:
>
> - We inject ***small random perturbations*** into the 3D coordinates and timestamps to mimic coordinate deviations and temporal misalignment.
> - We fine-tune LLaVA-4D on this perturbed version of Chat4D and compare the result to the original model.
>
> The performance is:
>
> |             Fine-tuning strategy              |      C$\uparrow$       |     B-4$\uparrow$      |     SAcc$\uparrow$     |     TAcc$\uparrow$     |
> | :-------------------------------------------: | :--------------------: | :--------------------: | :--------------------: | :--------------------: |
> |               Our Base LLaVA-4D               |          93.5          |          17.2          |          58.9          |          54.6          |
> | + Fine-tuned on Chat4D with annotation errors | 93.0 (0.5$\downarrow$) | 17.0 (0.2$\downarrow$) | 58.1 (0.8$\downarrow$) | 53.9 (0.7$\downarrow$) |
>
> The drops across all metrics are ***within 1 point***, which is very small relative to the absolute performance level. This suggests that:
>
> - Our LLaVA-4D is ***robust to minor spatial and temporal noise*** in the annotations.
> - The model focuses on ***cross-modal alignment patterns*** instead of memorizing exact coordinates or timestamps, thus small local errors do not significantly affect instruction tuning.
>
> In summary:
>
> - The core 3D and temporal labels come from detection and video processing, then pass through automatic and manual cleaning tailored to remove coordinate and timestamp errors.
> - Additional experiments with synthetic perturbations show that LLaVA-4D maintains almost the same performance, even in the presence of controlled label noise.

---

> > ### Comment · Reviewer_n3mW · 2025-11-24
> >
> > Thanks for your detailed rebuttals. Although the effective of data cleaning and the robustness of LLaVA-4D is clear, it remains two minor details to be clarified.
> >
> > 1. The quantitative results of Data Cleaning. How much data would be removed or repaired during the cleaning phrase? It's interesting to find out the accuracy of the process of data annotation.
> >
> > 2.  The details of small random perturbations, like the distribution and intensity.

---

> > > ### Author Response · Authors · 2025-11-24
> > > **Rebuttal to Two Minor Details**
> > >
> > > Thanks for the reviewer’s comments. We provide additional details on the data cleaning and the small perturbation, and have updated the corresponding descriptions in the revised manuscript accordingly.
> > >
> > > **Q1:** **[Details of Data Cleaning]**. The quantitative results of Data Cleaning. How much data would be removed or repaired during the cleaning phrase? It's interesting to find out the accuracy of the process of data annotation.
> > >
> > > **A:** As previously described, our data cleaning process consists of automatic filtering and manual inspection. The detailed statistics are as follows:
> > >
> > > - **Automatic filtering stage.** The original dataset is treated as 100%, and approximately **4.7%** of the samples are removed during automatic filtering. Specifically, we discard **2.4%** of samples with *inconsistent motion* in the 3D coordinates, **1.1%** of samples with *impossible trajectories* caused by temporal discontinuities, and **1.2%** of samples with *mismatched spatial regions*.
> > >
> > > - **Manual inspection stage.** After automatic filtering stage, we further remove approximately **0.8%** of the remaining samples through manual inspection. Specifically, we discard **0.3%** of samples with *minor timestamp misalignments*, **0.4%** of samples with *incorrect color or attribute descriptions*, and **0.1%** of samples with *large coordinate deviations* that were not detected during automatic filtering.
> > >
> > > After these two rounds of data cleaning, a total of approximately **5.5%** of the samples are removed, and more than **94%** of the dataset is retained as high-quality data.
> > >
> > > **Q2:** **[Details of Small Random Perturbations]**. The details of small random perturbations, like the distribution and intensity.
> > >
> > > **A:** To evaluate the robustness of our model to annotation errors, we introduce small random perturbations by adding mild Gaussian noise to both the 3D coordinates and the timestamps. The details are as follows:
> > >
> > > **1) Spatial noise.** We inject independent Gaussian perturbations into 3D coordinates:
> > > $$
> > > \tilde{x} = x + \epsilon_x \sim \mathcal{N}(0, \sigma_x^2), \quad \sigma_x = 0.02-0.05 ~m
> > > $$
> > >
> > > This denotes a spatial deviation of 2–5 cm. Such a magnitude is similar to the typical accuracy of mainstream RGB-D or monocular 3D object detection models, making it a reasonable and realistic error range.
> > >
> > > **2) Temporal noise.** We add Gaussian noise directly to the frame timestamps and clip the resulting perturbation within $\pm$2 frames:
> > > $$
> > > \tilde{t} = t + \epsilon_t \sim \mathcal{N}(0, \sigma_t^2), \quad \sigma_t = 0.5 ~ frame, \quad |\epsilon_t| \leq 2 ~ frame
> > > $$
> > > The noise is added directly to the timestamps, resulting in a final temporal shift within $\pm$2 frames. This range is consistent with the small frame misalignments commonly observed in video parsing.
> > >
> > > In summary, we present the distribution and intensity of the small random perturbations as follows:
> > >
> > > | Annotation Type | Noise Injection      | Distribution       | Intensity                                  |
> > > | --------------- | -------------------- | ------------------ | ------------------------------------------ |
> > > | 3D Coordinate   | Added to coordinates | Gaussian           | $\sigma$=2-5 cm                            |
> > > | 1D Timestamp    | Added to timestamps  | Gaussian (clipped) | $\sigma$=0.5 frame, capped at $\pm$2 frame |

---

> > > > ### Comment · Reviewer_n3mW · 2025-11-25
> > > >
> > > > Thanks for your comments. My concerns have been addressed and all the results are positive. But I'm unfamiliar with the 4D part, I would like to keep my score.

---

> > > > > ### Author Response · Authors · 2025-11-26
> > > > >
> > > > > We are pleasure that your concerns are addressed. Thanks for your valuable comments.

---

> ### Author Response · Authors · 2025-11-20
> **Rebuttal to Q2 and Q3**
>
> **Q2:** **[Effect of Optical Flow]** Temporal coding: it relies solely on optical flow to estimate motion information. However, optical flow is prone to inaccurate estimation in fast-moving or occluded scenes, which may lead to deviations in the temporal characteristics of dynamic objects and affect the accuracy of the model's trajectory prediction for high-speed moving objects.
>
> **A:** In our model, optical flow is used only as an **auxiliary** motion cue for temporal encoding instead of as the sole source of temporal information for dynamic scene understanding. We also model time through temporal disentanglement of visual features, time embeddings, and cross-attention, which together strengthen temporal reasoning in dynamic scenes.
>
> As shown in Table 5 of the supplementary material, we evaluate the model with and without optical flow in the coordinate encoding. Optical flow brings a clear but moderate improvement, and the model still achieves strong spatiotemporal understanding even without it. Furthermore, Table 11 reports results on fast-motion scenarios. The model remains robust to rapidly moving objects with only slight performance degradation.
>
> Therefore, the impact of optical flow errors on our performance is limited. Multimodal fusion and contextual reasoning in the LLM help compensate for local temporal misalignments introduced by imperfect flow.
>
>
>
>
>
> **Q3:** **[Temporal Coding]** Lack of experiments: the contribution in TAcc of optical flow and the kind of temporal coding like frame rate-based vs. motion speed-based are unclear.
>
> **A:** The functional roles of optical flow and temporal encoding are distinct. Optical flow provides auxiliary local motion features to assist temporal encoding, while temporal encoding is responsible for modeling global temporal relationships.
>
> **1) Regarding the contribution of optical flow**, we have included an analysis experiment in Table 5 of the supplementary material, which examines its impact on model performance. The results confirm that incorporating optical flow significantly enhances the ability of our model to understand independent dynamic patterns, particularly improving the TAcc metric.
>
> **2) Regarding the comparison of temporal encoding strategies**, we follow the suggestion raised by the reviewer to evaluate a frame rate-based temporal encoding scheme against our original motion speed-based (flow-based) encoding scheme. The results are shown in the following table:
>
> |      Temporal encoding      | C$\uparrow$ | B-4$\uparrow$ | SAcc$\uparrow$ | TAcc$\uparrow$ |
> | :-------------------------: | :---------: | :-----------: | :------------: | :------------: |
> |  Frame rate-based encoding  |    91.0     |     16.7      |      57.3      |      49.5      |
> | Motion speed-based encoding |  **93.5**   |   **17.2**    |    **58.9**    |    **54.6**    |
>
> We can observe that frame rate-based encoding performs worse than the motion speed-based encoding, especially in fine-grained temporal understanding. The underlying reason is that the frame rate-based approach uses a globally fixed-step encoding, which struggles to explicitly capture the temporal characteristics of independently moving objects in dynamic scenes. In contrast, the motion speed-based approach is a locally adaptive encoding scheme that is capable of directly representing the magnitude of motion, and thus reflecting the true physical dynamics of the scene.

---

> ### Author Response · Authors · 2025-11-20
> **Rebuttal to Q4, Q5 and Q6**
>
> **Q4:** **[Technical Details]** What is the initial of $w_p$ and $\beta$ ?
>
> **A:** Combining Equation (1) with the formula at Line 261 in the main text, we can see that $w_p$ is essentially a multi-layer perceptron (MLP) with small-variance normal initialization, used to align feature dimensions. Similarly, by comparing Equation (1) and Equation (6) in the main text,  $\beta$ can be interpreted as a motion speed embedding initialized with optical flow, which provides local dynamic information for temporal encoding. In the revised manuscript, we unify the variable notations to avoid potential misunderstandings.
>
>
>
> **Q5:** **[Technical Details]** The paper uses CLIP-ViT-L-336px as the visual encoder, but why not use a more suitable encoder for video (such as the Video Swin Transformer) was not chosen?
>
> **A:** It is important to emphasize that our goal is to build a general vision–language model with 4D awareness to enable multimodal reasoning for 4D dynamic scene understanding. Thus, the key requirement of the visual encoder is to possess cross-modal alignment capability for 4D scenes. Regarding the choice of the visual encoder, there is a fundamental difference between CLIP and Video Swin Transformer (VST) in both their modeling objectives and task orientations.
>
> - In terms of modeling objective, CLIP aims to learn semantic alignment between visual and linguistic modalities, while VST focuses on spatiotemporal modeling within the visual modality alone.
>
> - In terms of task orientation, CLIP is typically used for image understanding, visual question answering, and visual grounding, whereas VST is mainly applied to action recognition and video generation tasks.
>
> In contrast to CLIP that only requires the introduction of an effective 4D spatiotemporal embedding, VST additionally needs to learn cross-modal semantic alignment. Therefore, adopting CLIP serves as the optimal choice in the current version of our model to efficiently demonstrate the feasibility of this work. In the future, we plan to explore more video-based Transformer architectures as visual encoders to further enhance model performance.
>
>
>
> **Q6:** **[Model Differences]** What are the advantages of LMMs frameworks over pure vision models in 4D tasks, and how do they compare to 4D Gaussians and VG4D?
>
> **A:** In the table below, we list the differences between LMMs and pure vision models on 4D tasks, where LMMs are represented by our LLaVA-4D, and pure vision models are represented by 4D Gaussian [2] and VG4D [3].
>
> |        4D models        |     Example     |                  Output forms                  |   Modeling objectives    |            Learning hierarchy            |     Generalization     |
> | :---------------------: | :-------------: | :--------------------------------------------: | :----------------------: | :--------------------------------------: | :--------------------: |
> |   Pure vision models    |   4DGS, VG4D    |  Reconstruction, classification, recognition   |  Pixel-level estimation  | Pixel variations, geometric deformations | Task-specific training |
> | Large multimodal models | LLaVA-4D (Ours) | Semantic reasoning, instruction following, VQA | Semantic-level alignment | Causal relationships of dynamic patterns |   Zero-shot transfer   |
>
> We can observe the significant differences:
>
> - From the perspective of output forms, LMMs are capable of performing semantic reasoning, instruction following, and question answering, while 4DGS focuses solely on 4D scene reconstruction, and VG4D is oriented toward scene classification and recognition.
> - From the perspective of modeling objectives, LMMs aim to learn semantic-level cross-modal spatiotemporal alignment, and pure vision models focus on pixel-level geometric estimation.
> - From the perspective of learning hierarchy, LMMs can understand the causal relationships of dynamic patterns, and pure vision models only perceive pixel variations or geometric deformations.
> - From the perspective of generalization, LMMs exhibit strong zero-shot transfer capability, and pure vision models require task-specific training for each subtask, especially in the case of 4D Gaussian models.
>
> In summary, LMMs are more general and versatile for multi-task modeling in 4D scene understanding, while pure vision models are better suited for achieving peak performance on a single specific 4D task.

---

> ### Author Response · Authors · 2025-11-20
> **Reference**
>
> **Reference**
>
> [1] Rukhovich D, Vorontsova A, Konushin A. Imvoxelnet: Image to voxels projection for monocular and multi-view general-purpose 3d object detection. Proceedings of the IEEE/CVF winter conference on applications of computer vision. 2022: 2397-2406.
>
> [2] Wu G, Yi T, Fang J, et al. 4d gaussian splatting for real-time dynamic scene rendering. Proceedings of the IEEE/CVF conference on computer vision and pattern recognition. 2024: 20310-20320.
>
> [3] Deng Z, Li X, Li X, et al. Vg4d: Vision-language model goes 4d video recognition. 2024 IEEE International Conference on Robotics and Automation (ICRA). IEEE, 2024: 5014-5020.

---

### Official Review · Reviewer_gcu5 · 2025-11-01

**Soundness:** 3
**Presentation:** 3
**Contribution:** 2
**Rating:** 4
**Confidence:** 4

**Summary:**

This paper introduces LLaVA-4D, a framework that enhances MLLM for 4D spatiotemporal scene understanding with a new 4D feature embedding solution. The contribution is a spatiotemporal prompt that encodes 3D position and 1D time into a 4D coordinate embedding to better distinguish static backgrounds from dynamic objects. The model also disentangles visual features from multi-view videos into separate spatial and temporal components, which are then fused with the 4D prompts. Experiments are conducted on several 3D benchmarks and a new Chat4D benchmark.

**Strengths:**

- The paper presents a new 4D spatial-temporal understanding framework for MLLMs. Some new designs for visual encoding and positional embedding are proposed. A new data recipe and a new benchmark for 4D understanding are constructed.

- The new designs looks reasonable overall. Ablation studies show the effectiveness compared to baseline solutions.

**Weaknesses:**

- My main concern is the performance of the proposed method. A quite complex solution is proposed in the paper (with a new visual encoding solution and some new designs for embedding), but the results are not that impressive. Many recent methods, like Spatial MLLM [r1], 3UR-LLM [r2], and Coarse Correspondences [r3], that can achieve better performance on ScanQA are not compared or discussed.

[r1] Spatial-MLLM: Boosting MLLM Capabilities in Visual-based Spatial Intelligence

[r2] 3UR-LLM: An End-to-End Multimodal Large Language Model for 3D Scene Understanding, TMM

[r3] COARSE CORRESPONDENCES Boost Spatial-Temporal Reasoning in Multimodal Language Model, CVPR 2025

- How about the performance of LLaVA-4D on some new and more challenging benchmarks like VSI-Bench, which is specifically designed for 3D MLLM?

- For baselines of the new Chat4D benchmark, I would recommend adding results of SoTA proprietary models like GPT5, Gemini-2.5-pro, and open-source generalist MLLM like Qwen3-VL for better references. It would also provide more comparsions and discussions of the new benchmarks compared recent 3D/4D MLLM benchmarks. It would also be helpful to highlight the value and unqiue properties of the new benchmarks.

**Questions:**

Please refer to my comments above.

---

> ### Author Response · Authors · 2025-11-20
> **Strength Summary and Rebuttal to Q1**
>
> Thanks for affirming our contributions: **new MLLM framework** for 4D spatiotemporal understanding, **reasonable design** of visual encoding and positional embedding, **new 4D understanding dataset**, **ablation studies** demonstrating the model effectiveness.
>
> As for the raised issues, we summarize them into three aspects: more comparison with recent LMMs, additional evaluation, dataset value. Next, we address these issues one by one.
>
>
>
> **Q1:** **[Model and Performance]** My main concern is the performance of the proposed method. A quite complex solution is proposed in the paper (with a new visual encoding solution and some new designs for embedding), but the results are not that impressive. Many recent methods, like Spatial MLLM [1], 3UR-LLM [2], and Coarse Correspondences [3], that can achieve better performance on ScanQA are not compared or discussed.
>
> **A:** Thank you for raising this concern about model complexity and performance. Our main goal is to design a *general* model for 4D dynamic scene understanding, and our contributions lie in both the architecture and its strong performance on 4D benchmarks.
>
> **1) Model design: conceptually simple, modular 4D extension**
>
> Although the framework may appear complex at first glance, it decomposes cleanly into two standard, modular parts:
>
> - **4D visual representation.** We start from conventional visual encoders. We then construct ***spatial*** and ***temporal*** features and inject 4D awareness into the representation. These additions are minimal but necessary to capture object motion and temporal context in dynamic scenes.
> - **4D language representation.** We adopt a similar idea for text with position and time embeddings for language tokens. This design promotes fine-grained spatiotemporal alignment between language queries and 4D visual features.
>
> Each component follows standard practice in multimodal modeling. We ***do not*** introduce ad-hoc branches or task-specific tricks. Instead, we extend 2D/3D MLLM design principles into 4D in a unified and end-to-end framework that remains structurally simple once separated into these two parts.
>
> **2) Performance: modest 3D gains, large 4D gains, and new comparisons on ScanQA**
>
> Regarding performance, we distinguish between 3D and 4D settings:
>
> - **3D scene understanding.** As reported in Table 1 of the main paper, and Tables 13–14 and Figures 1–3 of our Supplementary Material, our method achieves *state-of-the-art* or near state-of-the-art performance on the 3D scene understanding benchmark with consistent but moderate gains. This suggests that our 4D extensions ***do not harm standard 3D capability*** and already provide some benefit.
> - **4D scene understanding.** On 4D benchmarks, our model yields a ***clear performance leap***, especially on fine-grained metrics that require temporal reasoning and dynamic scene understanding. This is where our design is intended to shine, and the results match that intention.
>
> To directly address the reviewer’s point on missing baselines, we further evaluate our model on ScanQA [4] and compare it with several recent 3D MLLMs: Spatial-MLLM [1], 3UR-LLM [2], and GPT-4o with Coarse Correspondences [3]. The results are:
>
> |              Method              | C$\uparrow$ | B-4$\uparrow$ | M$\uparrow$ | R$\uparrow$ |
> | :------------------------------: | :---------: | :-----------: | :---------: | :---------: |
> |           Spatial-MLLM           |    91.8     |     14.8      |    18.4     |    45.0     |
> |             3UR-LLM              |    87.7     |     15.5      |    18.4     |    41.5     |
> | GPT-4o + Coarse  Correspondences |    87.0     |      --       |    18.0     |    42.6     |
> |               Ours               |  **97.8**   |   **17.9**    |  **21.2**   |  **51.8**   |
>
> These results show that our ***LLaVA-4D outperforms all three strong 3D baselines on ScanQA across all reported metrics***. We attribute the gains mainly to our multi-view spatial features and 4D-aware representation, which strengthen spatial reasoning and temporal grounding.
>
> Due to the lack of released fine-tuning code for Spatial-MLLM, and the absence of code for 3UR-LLM and Coarse Correspondences, we cannot re-train these models on our own Chat4D dataset. We will clarify this limitation and include the above ScanQA comparison table in the revised version.
>
> In summary:
>
> - Our architecture is modular and conceptually simple once decomposed into 4D visual and language branches.
> - Our model achieves ***state-of-the-art or competitive performance on 3D benchmarks*** *and* ***significant improvements on 4D tasks***, where temporal reasoning is essential.
> - New experiments on **ScanQA** show that our approach ***outperforms recent 3D MLLMs***.

---

> ### Author Response · Authors · 2025-11-20
> **Rebuttal to Q2**
>
> **Q2:** **[Additional Evaluation]** How about the performance of LLaVA-4D on some new and more challenging benchmarks like VSI-Bench, which is specifically designed for 3D MLLM?
>
> **A:** We appreciate the suggestions of the reviewer. We adopt VSI-Bench [5] as a new evaluation benchmark for 3D scene understanding, and compare our model with several representative LLMs: LLaVA-Video-7B [6], LLaVA-OneVision-7B [7], LongVA-7B [8], VILA-1.5-8B [9], InternVL2-8B [10], LongVILA-8B [11] and Spatial-MLLM [1].
>
> |       Method       | Average  | Numerical Answer |            |           |           | Multiple-Choice Answer |           |            |             |
> | :----------------: | :------: | :--------------: | :--------: | :-------: | :-------: | :--------------------: | :-------: | :--------: | :---------: |
> |                    |          |    Obj. Count    | Abs. Dist. | Obj. Size | Room Size |       Rel. Dist.       | Rel. Dir. | Route Plan | Appr. Order |
> |   LLaVA-Video-7B   |   35.6   |       48.5       |    14.0    |   47.8    |   24.2    |          43.5          |   42.4    |  **34.0**  |    30.6     |
> | LLaVA-OneVision-7B |   32.4   |       47.7       |    20.2    |   47.4    |   12.3    |          42.5          |   35.2    |    29.4    |    24.4     |
> |     LongVA-7B      |   29.2   |       38.0       |    16.6    |   38.9    |   22.2    |          33.1          |   43.3    |    25.4    |    15.7     |
> |    VILA-1.5-8B     |   28.9   |       17.4       |    21.8    |   50.3    |   18.8    |          32.1          |   34.8    |    31.0    |    24.8     |
> |    InternVL2-8B    |   37.5   |       31.3       |    29.0    |   48.9    |   44.2    |          38.0          |   33.4    |    28.9    |    46.4     |
> |    Spatial-MLLM    |   48.4   |       65.3       |    34.8    |   63.1    |   45.1    |          41.3          | **46.2**  |    33.5    |    46.3     |
> |  LLaVA-4D (Ours)   | **48.6** |     **68.2**     |  **35.3**  | **64.8**  | **49.6**  |        **44.5**        |   45.2    |    33.8    |  **47.5**   |
>
> The results show that our LLaVA-4D achieves the best performance on most metrics, with only a few metrics slightly lower than the competing methods. This also demonstrates the superiority of our model in multimodal spatial reasoning.

---

> ### Author Response · Authors · 2025-11-20
> **Rebuttal to Q3**
>
> **Q3:** **[Dataset Value]** For baselines of the new Chat4D benchmark, I would recommend adding results of SoTA proprietary models like GPT-5, Gemini-2.5-pro, and open-source generalist MLLM like Qwen3-VL for better references. It would also provide more comparisons and discussions of the new benchmarks compared recent 3D/4D MLLM benchmarks. It would also be helpful to highlight the value and unique properties of the new benchmarks.
>
> **A:** This is a valuable suggestion. We conduct both model-level and benchmark-level comparisons to highlight the value and uniqueness of our Chat4D benchmark. The former focuses on comparing our model with other strong baselines on the Chat4D dataset, while the latter analyzes the differences between Chat4D and existing scene understanding benchmarks.
>
> **1) For the model-level comparison,** we evaluate our model against representative 3D LMMs (*e.g.*, 3D-LLM [12], 3D-LLaVA [13], LLaVA-3D [14], Video-3D LLM [15]) and the latest 2D state-of-the-art LMMs (*e.g.*, GPT-5, Gemini-2.5-Pro [16], Qwen3-VL [17]) on our Chat4D benchmark. The specific results are shown in the following table:
>
> |     Method      | C$\uparrow$ | B-4$\uparrow$ | M$\uparrow$ | R$\uparrow$ | SAcc$\uparrow$ | TAcc$\uparrow$ |
> | :-------------: | :---------: | :-----------: | :---------: | :---------: | :------------: | :------------: |
> |     3D-LLM      |    61.6     |     11.5      |    12.3     |    33.5     |      31.4      |       --       |
> |    3D-LLaVA     |    85.1     |     16.0      |    18.2     |    44.2     |      52.0      |       --       |
> |    LLaVA-3D     |    87.4     |     14.8      |    19.4     |    47.5     |      45.6      |       --       |
> |  Video-3D LLM   |    89.4     |     16.1      |    19.2     |    48.3     |      52.8      |       --       |
> |     GPT-5*      |    34.1     |      8.2      |    11.6     |    20.2     |       --       |       --       |
> | Gemini-2.5-pro* |    38.5     |      9.8      |    10.9     |    21.7     |       --       |       --       |
> |    Qwen3-VL     |    63.8     |     12.0      |    12.5     |    31.4     |      20.3      |       --       |
> | LLaVA-4D (Ours) |  **93.5**   |   **17.2**    |  **21.0**   |  **50.2**   |    **58.9**    |    **54.6**    |
>
> ** indicates that the model cannot be fine-tuned.*
>
> The results show that our model significantly outperforms the competing methods on spatiotemporal understanding metrics, demonstrating the effectiveness of our model in achieving fine-grained spatiotemporal understanding of dynamic scenes after being trained on Chat4D.
>
> **2) For the benchmark-level comparison,** we further compare our Chat4D with other 3D benchmarks (*e.g.*, ScanQA [4] and ScanRef [18]) in terms of input modality, dimension, data source, task coverage, and annotation type. The detailed comparison is shown in the following table:
>
> |    Dataset    |      Modality      |  Dimension   |  Data source  |           Task coverage           |                       Annotation type                        |
> | :-----------: | :----------------: | :----------: | :-----------: | :-------------------------------: | :----------------------------------------------------------: |
> |    ScanQA     |    Image, text     |   3D only    | Static scene  |          Captioning, VQA          |                           QA pairs                           |
> |    ScanRef    |    Image, text     |   3D only    | Static scene  |         Visual grounding          |                     Object-level labels                      |
> | Chat4D (Ours) | Image, video, text | 4D (3D+time) | Dynamic scene | Captioning, VQA, visual grounding | Object-level & Scene-level labels , Temporal relations, QA pairs |
>
> Compared to existing 3D benchmarks, our Chat4D provides richer input modalities and more fine-grained spatiotemporal textual annotations for dynamic scenes.
>
> In summary, our dataset fills the gap between 3D and 4D understanding, and also enhances fine-grained spatiotemporal understanding capability of LMMs for 4D dynamic scenes.

---

> ### Author Response · Authors · 2025-11-20
> **Reference**
>
> **Reference**
>
> [1] Diankun Wu, Fangfu Liu, et al. Spatial-mllm: Boosting mllm capabilities in visual-based spatial intelligence. arXiv, 2025.
>
> [2] Haomiao Xiong, Yunzhi Zhuge, et al. 3ur-llm: An end-to-end multimodal large language model for 3d scene understanding. IEEE TMM, 2025.
>
> [3] Benlin Liu, Yuhao Wang, et al. Coarse Correspondences Boost Spatial-Temporal Reasoning in Multimodal Language Model. IEEE CVPR, 2025.
>
> [4] Daichi Azuma, Taiki Miyanishi, et al. Scanqa: 3d question answering for spatial scene understanding. IEEE CVPR, 2022.
>
> [5] Jihan Yang, Shusheng Yang, et al. Thinking in Space: How Multimodal Large Language Models See, Remember, and Recall Spaces. CVPR, 2025.
>
> [6] Yuanhan Zhang, Jinming Wu, et al. LLaVA-Video: Video Instruction Tuning With Synthetic Data. TMLR, 2025.
>
> [7] Bo Li, Yuanhan Zhang, et al. LLaVA-OneVision: Easy Visual Task Transfer. arXiv, 2024.
>
> [8] Peiyuan Zhang, Kaichen Zhang, et al. Long context transfer from language to vision. arXiv, 2024.
>
> [9] Ji Lin, Hongxu Yin, et al. Vila: On pre-training for visual language models. CVPR, 2024.
>
> [10] Zhe Chen, Weiyun Wang, et al. How far are we to gpt-4v? closing the gap to commercial multimodal models with opensource suites. arXiv, 2024.
>
> [11] Fuzhao Xue, Yukang Chen, et al. Longvila: Scaling long-context visual language models for long videos. arXiv, 2024.
>
> [12] Hong Y, Zhen H, Chen P, et al. 3d-llm: Injecting the 3d world into large language models. Advances in Neural Information Processing Systems, 2023, 36: 20482-20494.
>
> [13] Deng J, He T, Jiang L, et al. 3d-llava: Towards generalist 3d lmms with omni superpoint transformer. Proceedings of the Computer Vision and Pattern Recognition Conference. 2025: 3772-3782.
>
> [14] Zhu C, Wang T, Zhang W, et al. Llava-3d: A simple yet effective pathway to empowering lmms with 3d-awareness. arXiv preprint arXiv:2409.18125, 2024.
>
> [15] Zheng D, Huang S, Wang L. Video-3d llm: Learning position-aware video representation for 3d scene understanding. Proceedings of the Computer Vision and Pattern Recognition Conference. 2025: 8995-9006.
>
> [16] Comanici G, Bieber E, Schaekermann M, et al. Gemini 2.5: Pushing the frontier with advanced reasoning, multimodality, long context, and next generation agentic capabilities. arXiv preprint arXiv:2507.06261, 2025.
>
> [17] Yang A, Li A, Yang B, et al. Qwen3 technical report. arXiv preprint arXiv:2505.09388, 2025.
>
> [18] Chen D Z, Chang A X, Nießner M. Scanrefer: 3d object localization in rgb-d scans using natural language. European conference on computer vision. Cham: Springer International Publishing, 2020: 202-221.

---

> ### Comment · Reviewer_gcu5 · 2025-11-26
> **Thanks for your feedback**
>
> Thanks for the detailed reply. The rebuttal addressed some of my concerns about the performance and the new benchmark. Although the method cannot achieve state-of-the-art performance on 3D spatial understanding benchmarks like VSIBench, I think the contributions on 4D modeling and evaluation are valuable. Thus I would upgrade my rating. It would be better to add results of 3D/spatial MLLMs on the table for Q2 for references.

---

> > ### Author Response · Authors · 2025-11-26
> >
> > Thank you for affirming our 4D modeling and evaluation. We have also updated the evaluation results of the 3D multimodal large model (*e.g.*, Spatial-MLLM [1]) on the table for Q2, and more evaluation results of 3D models will be presented in the final revised manuscript.

---

### Author Response · Authors · 2025-11-22
**To All Reviewers**

We sincerely thank the Reviewer gcu5, Reviewer n3mW and Reviewer NHKE for their valuable reviews and constructive comments. We also thank them for affirming our main contributions: **new, groundbreaking and innovative** MLLM  framework with significant improvement [gcu5, n3mW, NHKE], **new and contributed** dataset that **advances future research** [gcu5, n3mW, NHKE], **reasonable and stronger** visual feature encoding and positional embedding [gcu5, NHKE], ablation studies [gcu5, NHKE], three-stage training strategy [n3mW], **clear writing** [n3mW], **easily-understood motivation** [n3mW].

Regarding the comments and suggestions raised by the reviewers, we have provided point-by-point answers. Moreover, we have carefully adopted their valuable suggestions in the revised manuscript in ***blue*** text.

---

### Meta-Review · Area_Chair_XQfS · 2025-12-11

**Summary:**

The authors innovatively proposed a multimodal large model for 4D scenes and constructed a large-scale dataset. Three reviewers rated this paper near the borderline but overall positively. The authors submitted a rebuttal, and two reviewers participated in the discussion, indicating that the authors addressed most of their concerns. The AC agreed to accept the paper, recommending that reviewers supplement 3D-related evaluation results and data cleaning details in subsequent stages, and open-source relevant code and data.

Additionally, the citation in this paper reads: "Mandy Chen, Adams Wei Yu, Hamid Palangi, Paul Smolensky, Yinfei Yang, Xiaowei Yuan, Kathy Meier-Hellstern, Jianfeng Gao, Ed Chi, et al. Pali: A jointly-scaled multilingual language-image model. arXiv preprint arXiv:2303.07892, 2023b." contains errors in the author sequence and arXiv ID. The authors are requested to correct these errors during the CA stage.

**Reviewer Concerns:**

Reviewers gcu5 and n3mW participated in the discussion, and most of their concerns have been addressed. Reviewer NHKE did not participate in the discussion. AC thoroughly reviewed this reviewer's questions and the authors' rebuttal, concluding that most of this reviewer's concerns have also been resolved.

**Reviewer Scores:**

AC believes that reviewer NHKE will maintain or increase his/her score after reading the author's rebuttal.

---

### Decision · Program_Chairs · 2026-01-26

Accept (Poster)